# Persistent Organic Pollutant Levels in Maternal and Cord Blood Plasma and Breast Milk: Results from the Rio Birth Cohort Pilot Study of Environmental Exposure and Childhood Development (PIPA Study)

**DOI:** 10.3390/ijerph20010778

**Published:** 2022-12-31

**Authors:** Aline Souza Espindola Santos, Josino Costa Moreira, Ana Cristina Simoes Rosa, Volney Magalhães Câmara, Antonio Azeredo, Carmen Ildes Rodrigues Froes Asmus, Armando Meyer

**Affiliations:** 1Occupational and Environmental Branch, Public Health Institute, Federal University of Rio de Janeiro, Rio de Janeiro 21941-598, Brazil; 2Center for Studies of Human Ecology and Worker’s Health, National School of Public Health, Oswaldo Cruz Foundation, Rio de Janeiro 21041-210, Brazil; 3School of Medicine, Maternity School Hospital, Federal University of Rio de Janeiro, Rio de Janeiro 22240-000, Brazil

**Keywords:** organochlorine compounds, DDT, polychlorinated biphenyls, pregnant women, umbilical cord, breast milk, Brazil

## Abstract

Levels of polychlorinated biphenyls (PCB) and organochlorine pesticides (OCP) were evaluated in the breast milk and maternal and umbilical cord blood of pregnant women and their newborns in Rio de Janeiro, Brazil. The concentration of 11 PCB and 17 OCP were measured in 135 samples of maternal, and 116 samples of cord blood plasma, as well as 40, 47, and 45 samples of breast milk at 1st, 3rd, and 6th months after birth, respectively, using gas chromatography-mass spectrometry (GC-MS-MS). Women were asked to answer an enrollment questionnaire with reproductive, lifestyle, residential and sociodemographic questions. The most commonly detected OCPs and PCBs in the maternal and cord blood were 4,4′-DDE; β-HCH; ɣ-HCH; and PCB 28. 4,4′-DDE was also the most commonly detected OCP in breast milk samples. Although not statistically significant, ∑DDT levels were higher among women with pregestational BMI ≥ 30, and who were non-white and older (age > 40). Newborns with an Apgar score ≤ 8 at minute 5 of life showed significantly higher levels of ∑DDT in the cord blood. Persistent OCPs and PCBs were still detected in maternal and umbilical cord blood and breast milk, even after decades of their banishment in Brazil. They may pose a risk to maternal, fetal and children’s health.

## 1. Introduction

Organochlorine pesticides (OCP) and polychlorinated biphenyls (PCB) are classified as environmentally persistent organic pollutants due to their physical, chemical and toxicological properties, and pose a considerable threat to human health [1]. PCBs were used as flame retardants in industrial consumer products and old electrical devices, especially transformers and capacitors, which remain the primary source of these pollutants in urban areas of Brazil and around the world [2]. Brazil has prohibited their commercialization and manufacture through the Inter-Ministerial Ordinance No. 19, of 29 January 1981, with plans to eliminate these contaminants by 2028. OCPs were largely used in Brazil between the 1960s and mid-1980s in agricultural and public health campaigns [3]. In 1985, the use of OCPs as agricultural pesticides was prohibited in Brazil but allowed for other purposes, such as household pesticides, medicine to treat lice, wood preservatives and control of vector-borne diseases, until the early 2000s [3,4].

Despite this long-term prohibition, OCP and PCB residues are still found in the environment and food chain [5], representing Brazil’s main pathway for human exposure [4]. Although OCP and PCB were banned in Brazil about 30–40 years ago, these substances are ubiquitous and stable; therefore, their reduction and elimination from the environment or food chain are prolonged. Several OCP, especially DDT metabolites, are reportedly found in highly consumed fish collected at Guanabara Bay, Rio de Janeiro, Brazil [6]. In addition, some of the pollutants found in Rio de Janeiro City air samples were DDT (median = 233 pg/m^3^) and HCH (median = 340 pg/m^3^) [7].

Due to their persistence and potential carcinogenicity, immunotoxicity, reproductive toxicity, neurotoxicity and endocrine-disrupting effects, OCP and PCB represent significant public health concerns [8]. These substances can cross the placental barrier and endanger the fetus’ health [9]. Exposure to OCP and PCB during the first years of life is also impactful because the nervous and immune systems, as well as xenobiotic metabolic pathways, continue to develop until early childhood [10]. Studies have reported the presence of PCB and OCP or their metabolites in the blood of pregnant women [11], umbilical cord blood [12] and breast milk [13]. Other studies suggested that prenatal exposure to these substances is associated with adverse birth outcomes such as reduced size, weight and head circumference [14], and delayed neurodevelopment [15]. Some undesirable health effects have been found in descendants even after their grandmother’s exposure to DDT [16].

In Brazil, only a few studies have examined the concentration of persistent organic pollutants in maternal and umbilical cord blood. We investigated the levels of 11 PCB and 17 OCP in the breast milk and blood of pregnant women, and their newborns’ umbilical cords, enrolled in the pilot study of the Rio Birth Cohort (PIPA Study), in the city of Rio de Janeiro, Brazil. We also explored the relationship between the sociodemographic characteristics of the mothers and certain birth outcomes with the concentrations of the studied OCP and PCB.

## 2. Materials and Methods

### 2.1. Participants and Enrollment

PIPA is a hospital-based birth cohort study designed to examine the relationship between environmental exposure to several pollutants during pregnancy and any adverse effects upon delivery and the physiological development of the newborn from infancy to the age of 4. A pilot study was conducted from September 2017 through February 2018 and enrolled 142 pregnant women (Figure 1). In the current study, we described the levels of OCPs and PCBs in the maternal and umbilical cord blood of these women and their newborns [17].

Pregnant women over 16 years of age and at the beginning of the third trimester of pregnancy were invited to participate in the Rio Birth Cohort pilot study during their visit to the Federal University of Rio de Janeiro’s Maternity Hospital (UMH) in October and November of 2017. This hospital serves women living in low-income communities in the southern part of Rio de Janeiro City. High-risk pregnancies identified at any other medical unit in Rio de Janeiro may also be referred to the UMH.

### 2.2. Sampling

Out of 209 eligible pregnant women, 142 (67.9%) accepted the invitation to participate in the PIPA pilot study. A questionnaire containing socio-demographics, lifestyle (i.e., smoking, alcohol, physical activity), and prenatal questions was completed by all participants, but only 139 women (98%) authorized the use of blood samples for chemical analysis. Four samples showed insufficient amounts and were discarded.

A total of 135 deliveries occurred at UMH from October 2017 through February 2018. Samples of umbilical cord blood were collected from 126 (93%) of the newborns. The OCP were quantified in 135 (95%) maternal blood and 116 (86%) umbilical cord blood samples, as 10 of these samples showed hemolysis. Breast milk samples were collected from 40, 47, and 45 mothers at the 1st, 3rd, and 6th months of follow-up.

### 2.3. Analytical Methods

The OCP and PCB were analyzed through GC-MS/MS and included isomers of aldrin, dieldrin, endrin, DDT, DDD, DDE, HCH, HCB, mirex, endosulfan, chlordane, dicofol, heptachlor, methoxychlor, nonachlor pentachloroanisole, and the PCB congeners 28, 31, 52, 77, 101, 105, 118, 126, 128, 138, 153, 156, 169, 170 and 180. The analytical method has been adapted from Sarcinelli et al. [18].

Briefly, plasma samples were allowed to equilibrate at room temperature, and subsequently denatured and diluted with equal parts of methanol and water, then mixed and extracted on C18 solid-phase extraction (SPE) cartridges (JT Baker, Phillipsburg, NJ, USA). Each cartridge was dried and eluted with 7 mL of hexane. The eluate was applied to a florisil SPE cartridge, then eluted with petroleum ether: hexane (85:15) solution. The extracts were evaporated under a nitrogen atmosphere, and the volume was resuspended to 100 µL with hexane and analyzed by GC-MS/MS using 5 µL of 1,1’-biphenyl-4,4′dibromine at 1 µg mL^−1^ as the internal standard.

The GC-MS/MS analysis was performed using a Thermo Scientific, Waltham, MA, USA model TSQ 8000 EVO Pesticide Analyzer equipped with a Trace GC 1310 and an AS 1310 autosampler with a programmable temperature vaporization injector operating in the splitless mode using Thermo Fisher Xcalibur™ and TraceFinder™ software V.5. An Agilent DB-5MS phenylmethyl siloxane (30 m × 250 μm × 0.25 μm) column was used with ultrapure helium as the carrier gas at a constant flow rate of 1 mL min^−1^. The injector temperature was 280 °C, and splitless injection occurred for 1 min. The flow was purged at 30 mL min^−1^ for 1.2 min and 2 μL of the sample was injected. The GC oven temperature ramp program mode was 50 °C (2 min) at 10 °C min^−1^ to 180 °C (0 min) at 3 °C min^−1^ to 230 °C (0 min) at 5 °C min^−1^ to 280 °C (0 min), and at 15 °C min^−1^ to 310 °C (7 min), for a total of 50.68 min. The detector temperature was set at 300 °C. Compounds were identified by selected reaction monitoring (SRM) adjusted for retention time. Spectrometric conditions were defined according to the National Institute of Standards and Technology (NIST) libraries included in the Tracefinder™ software, although some confirmations using Thermo AutoSRM were necessary; thus, the transitions were defined according to their high specificity combined with high abundance.

OCP concentrations were adjusted by concentrations of lipids in breast milk. For this, 9 mL of a mixture of hexane-acetone was added to a 1 mL milk sample. The mixture was vortexed for 1 min, then placed in ultrasound for 20 min and centrifuged at 4000 rpm ± 100 rpm for 15 min. The supernatant in previously weighed tubes was collected. The extraction was repeated. The supernatant solvent was evaporated under an N2 atmosphere until dry. The total fat was calculated by the weight difference of the tubes.

### 2.4. Quality Assurance and Quality Control for OCP and PCB Analysis

Spiked plasma were used to prepare calibration curves. Validation parameters (Appendix A) included linearity, sensitivity, recovery tests, repeatability, the limit of detection (LOD) and limit of quantification (LOQ) based on the EPA 8081B method [19] and the Brazilian INMETRO Guide [20]. Eight-point calibration curves from 0.2 to 15 ng mL^−1^ were used to prepare the calibration curves by linear regression. Correlation coefficients ranged from 0.936 to 0.994, showing good linearity for both OCPs pesticides and PCBs. Sensitivity was calculated using the curve slope and varied from 111.655 to 212.082 for DDT and its isomers. The mean percentage recoveries ranged from 93% to 105% for OCP and from 87% to 107% for PCB, evaluated at 0.5, 5, and 15 ng mL^−1^. LOD and LOQ were calculated from standard deviation multiplied by factor t (Student t) or per 10, respectively, for 7 replicates at 0.2 ng mL^−1^ lowest level. The factor t for 6 degrees of freedom at a 95% confidence interval was 1.943. The LOD was 0.015 ng mL^−1^ to 0.468 ng mL^−1^ for OCP and 0.2 ng mL^−1^ to 0.36 ng mL^−1^ for PCB. The LOQ was 0.045 ng mL^−1^ to 1.419 ng mL^−1^ for OCP pesticides and 0.05 ng mL^−1^ to 1.08 ng mL^−1^ for PCB. Repeatability at 2 ng mL^−1^ level ranged from 0.5 to 14.9% for within-day variability and from 0.8 to 8.6% for day-to-day variability.

To perform only one analysis per sample and obtain one chromatogram and SRM spectrum, analytical quality controls were performed for each batch using previously spiked plasma at 3 levels: 0.2, 0.5, and 1 ng mL^−1^ and blank. The control of sample results was made by monitoring the daily abundance values in spiked plasma at these 3 concentration levels and the slope for each batch control. If the variation between the last batches was greater than 20%, the batch was redone. Additionally, variations were <20% for two randomized samples, with extra available volume, done in duplicate in each batch.

The internal standard was added in all samples and controls to evaluate the injection and chromatographic batch conditions, being monitored over time by its area, and all analytes in a standard mixture with greater purity than 99%, at 1 µg mL^−1^ level. Solvent injections were also made every 10 samples to evaluate the carry-over effect and cleanness of the chromatographic system.

Three selective reaction monitoring transitions were exhaustively checked for each positive sample to avoid false positive results. The relative abundances of the three selected precursor ion–product ion transitions were obtained from high purity standards and monitored for each pesticide. Additionally, a very low variation in retention times was observed. Finally, each batch had its blank, quality control samples and a short calibration curve in spiked plasma, and sample levels were calculated by external standardization using linear regression.

### 2.5. Estimated Daily Intake and Hazard Quotient Calculations for Children

The EDI was estimated using the following equation: EDI = Cmilk × Cfat × Mingestion/BW; where EDI = estimated daily intake; Cmilk = pesticide concentration of pesticide in milk (ng/g of lipid); Cfat = milk fat content (%) in breastmilk; Mingestion = intake of breast milk [21]. The adopted values for EDI estimation were: 4% of lipid [22] and 700 g of human milk/day [23]. The HQ was calculated as the EDI divided by the reference dose (RfD) proposed by EPA (5 μg of ΣDDT/kg body wt./day) [24] and the provisional tolerable weekly intake (PTWI) adopted by FAO/WHO (10 μg of ΣDDT/kg body wt./day) [25,26,27]. HQ values higher than 1 mean that there is a risk associated with breast milk consumption.

### 2.6. Statistical Analysis

Descriptive analysis included frequency distribution of maternal characteristics such as age groups (16–19, 20–39, ≥40 years), ethnicity (white, nonwhite), monthly household income (R$ ≤1733.33, 1733.34–3000.00, and >3000.00), schooling (high school or less, higher education), BMI (<25, 25–29.9, ≥30), alcohol consumption (yes or no), smoking (never, before pregnancy, during pregnancy), exposure to second-hand smoke (yes or no) and gestational age (<37 or ≥37 weeks). 

Newborn characteristics included sex, birth weight (<2500 or ≥2500 g), size for gestational age (small, appropriate, or large for gestational age) and Apgar at minute 5 (≤8 or >8). 

The concentration of DDT and its metabolites, HCH isomers, all other OCPs and PCBs in the maternal and umbilical cord blood samples were described using the following parameters: LOD in ng/mL, distribution of frequencies, percent above LOD, geometric mean (GM), standard deviation, minimum, maximum and the 25th, 50th and 75th percentiles. In addition to the 11 PCBs and 17 OCPs, we also calculated these parameters for the sum of the concentrations of all persistent organic pollutants studied (OCPs + PCBs = ∑POP), OCPs (∑OCPs), PCBs (∑PCB), HCH isomers (∑HCH) and DDT isomers/metabolites (ΣDDT). Blood concentrations below the LOD were excluded from the data set.

Frequency, GM, and its 95% confidence interval (95%CI) were also described for the sum of the compounds according to the mother’s demographic variables and newborn birth characteristics. Since only 4,4′-DDE had a detection rate above 5% in the breast milk samples, only its GM was described in the 1st, 3rd and 6th follow-up months. We also described the 4,4′-DDE breast milk concentration for paired samples of mothers during at least two follow-ups of 3. Spearman correlation between DDE, ΣHCH, ΣDDT, ΣOCP and ΣOC in the maternal and cord blood and breast milk during the 1° and 3° months of follow-up were calculated.

Predictors of ∑DDT levels in maternal blood were estimated by linear regression models, crude and adjusted. Maternal age, BMI, education and ethnicity were used as independent variables, while the maternal blood levels of ∑DDT was the dependent one. We also used linear regression models to evaluated the association between ∑DDT levels in umbilical blood (independent variable) and birth outcomes, such as birth weight and length and Apgar score (dependent variables).

## 3. Results

The main characteristics of the sample population are presented in Table 1. The majority of pregnant women (82.7%) were between 20 and 39 years of age and non-white (73.6%). Approximately 45.2% reported an average household income between R$ 1733 and 3000 (USD 306–530), but 36% reported it to be below R$ 1733. Most of the participants (76.3%) had a high school degree or less. Regarding pre-gestational BMI, 47% of the mothers were normal weight, 33.3% were overweight, and 19.7% were obese. Alcohol consumption during pregnancy was reported by 46.6% of the women, while 10.9% reported smoking before pregnancy, and 9.3% smoked during pregnancy. In addition, 48.4% reported living with a person who smoked. Among the newborns, males were slightly more frequent (54.9%) than females. Seven percent of newborns had low birth weights (<2500 g), 9.8% of newborns were delivered before the 37th week of gestation and 19.5% showed an Apgar score below 8. Newborns appropriate, large, and small for gestational age accounted, respectively, for 81.5%, 10.7%, and 7.8% of the sample.

The concentrations of PCBs and OCPs in maternal and umbilical cord blood samples are shown in Table 2. The most abundant specific OCP compounds in maternal blood (in descending order) were 4,4′-DDE (22.2%), β-HCH (8.89%), and ɣ-HCH (7.41%), while in the umbilical cord blood samples, the most frequently detected substances were 4,4′-DDE (15.5%), beta-HCH (11.2%), 4,4′-DDT (10.3%), and 4,4′-DDD (7.8%). The frequency of the summed variables was similar in both types of sample, with ∑DDT (33;24.4%), and ∑HCH (13; 9.6%) in maternal blood samples, and ∑DDT (33; 28.5%), and ∑HCH (15; 12.9%) in umbilical cord blood samples. The concentration GM and range of the most commonly detected OCPs in maternal blood were 0.050–3.196 ng/mL (GM = 0.131) for 4,4′-DDE; 0.090–0.179 ng/mL (GM = 0.121) for β-HCH; 0.145–0.251 ng/mL (GM = 0.175) for ɣ-HCH; 0.030–3.249 ng/mL (GM = 0.136) for ∑DDT; and 0.090–0.430 ng/mL (GM = 0.230) for ∑HCH. 

In the umbilical cord samples, these numbers were 0.052–5.440 ng/mL (GM = 0.150) for 4,4′-DDE, 0.023–0.169 ng/mL (GM = 0.071) for β-HCH, 0.050–0.477 ng/mL (GM = 0.099) for 4,4′-DDT, 0.030–0.226 ng/mL (GM = 0.066) for 4,4′-DDD, 0.030–6.231 ng/mL (GM = 0.151) for ∑DDT and 0.023–0.442 ng/mL (GM = 0.136) for ∑HCH (Table 2). The presence of 2,4′-DDT, 4,4′-DDD, 4,4′-DDT, dieldrin, endosulfan sulfate, methoxychlor, mirex, pentachloroanisole, and PCBs 28, 31, 52, 153 and 180 were detected in very few maternal blood samples (less than 5%), so their concentrations were not shown in the table. The same procedure was adopted for the description of concentrations of 2,4′-DDD, 2,4′-DDE, δ-HCH, endosulfan sulfate, dicofol, dieldrin, methoxychlor, mirex, pentachloroanisole and PCB 28, 31, 52, 101, 105, 118, 138, 128, 156 and 180 in umbilical cord blood samples. However, the levels of the compounds detected in less than 5% of the samples were included in ∑DDT, ∑HCH, ∑OCPs, and ∑POP.

Table 3 shows that, in general, the concentrations of ΣDDT, ΣHCH, ΣOCPs, and ΣPOP did not vary significantly as a function of maternal or newborns characteristics. Although not statistically significant, ∑DDT levels were slightly higher among non-white pregnant women, with an age > 40 and a pregestational BMI ≥ 30. On the other hand, newborns with an Apgar score of 8 or less showed significantly higher levels of ∑DDT in the umbilical cord blood.

The geometric mean of 4,4′-DDE concentrations in breast milk was lower in the 1st month of follow-up (n = 19; GM: 6.59; 95%CI: 3.33–13.05 ng/g) when compared with the 3rd (n = 22; GM: 15.12; 95%CI: 9.96–22.98 ng/g) and 6th (n = 6; GM: 11.41; 95%CI: 2.73–47.60 ng/g) months. ANOVA analysis showed no difference in 4,4′-DDE concentrations among the groups.

The calculated EDI varied from 5.98 × 10^−6^ to 1.17 × 10^−3^ μg of ΣDDT/kg body wt./day. The mean EDI was 0.21 μg of ΣDDT/kg body wt./day. The HQ values ranged from 5.98 × 10^−3^ and 1.20 × 10^−2^ to 1.17 and 2.34 according to EPA and FAO/WHO, respectively, and just one milk sample yielded an HQ value higher than 1.

Table 4 shows the correlation between DDE, ΣHCH, ΣDDT, ΣOCP, and ΣOC in the maternal and cord blood and breast milk during the 1° and 3° months of follow-up. Moderate correlations were observed between maternal ΣDDT and maternal ΣOCP (ρ = 0.709; *p* = <0.0001) and maternal ΣOC ρ = 0.684; *p* = <0.0001). ΣDDT in cord blood was most strongly correlated with ΣOCP (ρ = 0.913; *p* = <0.0001) and ΣOC (ρ = 0.911; *p* = <0.0001), whereas ΣOCP in cord blood was most strongly correlated with ΣOC ρ = 0.996; *p* = <0.0001) in cord blood. There were no correlations between DDE, ΣHCH, ΣDDT, ΣOCP, and ΣOC levels in the maternal and cord blood. Likewise, there were no correlations between DDE, ΣHCH, ΣDDT, ΣOCP and ΣOC levels in the maternal blood and breast milk.

Table 5 shows the results of the regression analysis for predictor factors of the ΣDDT levels in maternal blood. None of the analyses showed significant associations. Nevertheless, independently, age, BMI and being non-white showed trends in increasing maternal levels of ΣDDT, while having high school or higher education tended to decrease it. We also performed two adjusted models. The first one consisted of age as the main independent factor, adjusted for BMI. Although both variables still showed trends for increasing levels of maternal ΣDDT, their observed coefficients in the adjusted model were slightly lower than those in crude models. In the second adjusted model, we included ethnicity as an additional covariate. Age and BMI coefficients remained unchanged compared with model 1, and ethnicity’s coefficient was slightly higher than observed in the independent model.

Umbilical blood level of ΣDDT was associated with a significant decrease in the Apgar score at the 5th minute in the studied newborns. ∑DDT levels in the umbilical cord also showed trends in decreasing birth weight and gestational length; however, it was not statistically significant. When adjusted for birth weight (model 1), the association between umbilical blood ∑DDT and Apgar score at the 5th minute remained practically unchanged. Adding gestational length (model 2) led to a slight decrease in the β coefficient, but a significant confidence interval remained (Table 6).

## 4. Discussion

In this study, 4,4′-DDE, β-HCH, and ɣ-HCH were the most detected POPs in the blood of pregnant women; whereas, 4,4′-DDE; 4,4′-DDT and β-HCH were the most frequent OC detected in the umbilical cord blood of newborns. In a small number of cord blood samples, 4,4′DDT was identified. The presence of DDT in these samples can be related to the newborn difficulty in metabolizing 4,4′-DDT into 4,4′-DDE [28]. Although the presence and concentration of ∑DDT are decreasing with the age, some residues can be found even in some young mothers due to transference from parents and even grandparents [16]. In the present study, only one mother’s blood sample presented 4,4′-DDT and 4,4′-DDE simultaneously, and yielded a high ratio of 4,4′-DDT/4,4′-DDE (13.1). We also detected 4,4′-DDT in one breast milk sample. In this case, the 4,4′-DDT/4,4′-DDE ratio was 0.16. Mekonen and colleagues [21] observed higher 4,4′-DDT/4,4′-DDE ratios (varying from 0.66 to 0.81, according to specific locations) in Ethiopian breast milk samples than those observed in the present study. According to Cohn et al. [29], this ratio could indicate recent commercial DDT exposure. Organochlorine pesticides (α- and β-endosulfan; 4,4′ and 4,4′DDD; 2,4′ and 4,4′-DDE, 2,4′-DDT; α-, β-, δ- and γ-HCH; cis-chlordane and trans-chlordane, mirex, methoxychlor, dieldrin and heptachlor) were measured in polyurethane foam (PUF) disks used as passive air samplers in 10 sampling sites surrounding the Guanabara Bay [7]. Surfaces of particulate matter, fine soil particles and resuspended dust represent important environmental exposure sources of pollutants to humans and could be integrated by exposed PUF disks as well as semi-volatile forms of OC. These environmental compartments could be sampled using PUF by diffusion according to the second Fick’s Law of diffusion.

Our results demonstrated that elevated levels of ∑DDT were correlated positively with Apgar scores of 8 or less; while the mothers’ age, BMI and race showed no correlations. The influence of prenatal exposure to DDT and DDE in children’s neurodevelopmental development and the consequence of this exposure during childhood has been reported [15]. However, the influence of in utero exposure to DDT and DDE on Apgar scores is not fully understood.

Compared with similar studies, the concentration of 4,4′DDE found in this study was lower than that observed in pregnant women from Rio de Janeiro by Sarcinelli et al. [18], but similar to those reported by Bastos and colleagues [30]. However, the OCP concentrations reported by Sarcinelli et al. [18] were determined in blood samples collected from pregnant women between 1997 and 1998, when these substances were still being used in Brazil in campaigns for the control of vector-borne diseases. The study conducted by Bastos et al. [30] was conducted one decade later, after the prohibition of OCPs use in Brazil, and the women enrolled in their study were more likely exposed to levels of OCPs quite similar to those found currently. 

A comparison with studies conducted in other countries, that did not correct the OCPs concentrations by the total blood lipid content, showed that the concentrations of 4,4′-DDE observed in our study were lower than those reported in the Arctic region of Russia, Salta and Ushuaia localities (North and South regions, respectively), Argentina, and in healthy pregnant women from Algarve region, South Portugal [31,32,33]. However, levels of 4,4′-DDT and 4,4′-DDD in the umbilical cord blood samples were higher in our study than those observed in studies conducted in Spain [28]. Overall, these results may suggest that the exposure of pregnant women and their newborns to DDT may be declining. 

Similarly, in our study, β-HCH concentrations in maternal blood were lower than those observed in Argentina and Russia [31,32], but higher than those observed in Spain [28], while the concentration of ɣ-HCH was higher than that found in Russia [31]. In the umbilical cord blood samples, we observed β-HCH concentrations lower than those found in a study conducted in Canada [34], but higher than in Spain [28]. According to ATSDR [35], commercial lindane refers to products with more than 99% of ɣ-HCH. Among the HCH isomers, β-HCH presents slow-elimination kinetics and produces adverse effects such as red blood cell decrease, hemoglobin reduction and unconsciousness. 

Due to the abandonment of large amounts of technical-grade HCH waste in the metropolitan area of Rio de Janeiro in the 60s, non-occupational exposure to organochlorines has been frequently reported [36]. Additionally, OCPs were used in public health campaigns to control leishmaniasis in endemic areas of Rio de Janeiro City until the 1990s [37]. Although, the majority of studies conducted in Rio de Janeiro did not describe the location of the participants’ residences [4,30]. In our study, levels of ΣDDT, ΣHCH, ∑OCP, ∑POP, and ΣPCB did not differ statistically regarding the mother’s area of residence. These results may be due to the small number of samples with detectable levels of OCPs in this study. In the Rio Birth Cohort study, when 1000 pregnant women are expected to be enrolled, we may have enough statistical power to identify differences in organochlorine exposure based on the location of the mother’s residence, which may help to identify possible sources of organochlorine exposure in the city of Rio de Janeiro. 

In general, the concentrations of PCBs observed in the umbilical cord samples were similar to those observed in previous studies conducted in Rio de Janeiro; however, they were lower than those observed in a study conducted in the southern Brazilian city of Santa Maria [38]. PCB 138, 153, and 170 quantified in blood cord samples are congeners containing chlorine substitution at 2,2′ positions, an important chemical property that decreases the enzymatic metabolism rate once compared with non-2,2′ substituted PCB [38]. This characteristic also results in a higher trophic magnification observed in various studies worldwide. ATSDR [39] also points out that PCB 138 and 153 are the most prevalent congeners in human milk samples.

Socioeconomic and demographic characteristics have been positively correlated with OCP and PCB levels in maternal and umbilical cord blood in some epidemiological studies. For example, older pregnant women had the highest OCP concentration in several studies [31,32]. This study confirms this, as the level of DDT in blood, and its frequency, are proportional to age; older women (≥40 years) showed a higher incidence (77%) and higher levels of DDT in their blood. This result is expected because DDT was used extensively in Brazil until the 80s [4]. 

In the current study, participants with a BMI ≥ 30 also showed higher, though not statistically significant, concentrations of DDT in their blood. Few studies have evaluated robust associations between DDT and adiposity because the population’s detection rate is generally low. La Merril and colleagues [40] observed DDT and their metabolites summed were significantly associated with increased adiposity in confounder-adjusted models. This study’s detection rate for DDT and metabolites was above 90%. However, this association is inconsistent in birth cohort studies, and the differences in the detection rates among them may have impacted the estimates [41,42,43,44].

Generally, we observed that 4,4′-DDE concentration in breast milk in the 1st month was lower than in the 3rd and 6th-month follow-up. However, we observed an increase of 4,4′-DDE for 67% of mothers’ paired samples with at least two follow-ups of three. In the same way, Lakind et al. [45] observed that in paired samples of breast milk for months 1–3 postpartum, concentrations of 4,4′-DDE inconsistently decreased during lactation, and increased for 71% of women. Yu et al. [46] found no differences between organochlorine levels in colostrum (obtained until the fifth day postpartum) and those in mature milk samples (after 14 days). There is no consensus on the relationship between educational level, income and blood concentrations of organochlorines. Controversial associations have been reported for food intake, especially seafood and fish, and high income and educational levels [47,48]. According to the Brazilian Institute of Geography and Statistics [49], the average household income in Brazil between 2017–2018 was approximately USD 1000 monthly (R$ 5500/month). In the current study, the majority of the families received less than the Brazilian minimum salary (currently about R$ USD 200/month); so, at least 32% of the studied pregnant women were classified as poor or very poor. Our results showed that the ∑DDT was higher among women with a monthly income lower than R$ 1733 (USD 315), but this difference was not statistically significant.

DDT residues and HQ have been calculated in breast milk samples in Brazilian studies. For example, Sant’ana et al. [50] analyzed human milk samples collected from rural and urban areas in Botucatu City, Brazil. The values expressed in whole milk varied from 1.0 to 101.2 µg of total DDT (expressed as op’-DDT+pp’-DDT+pp’-DDD)/L, and from 1.0 to 62.1 µg of total DDT (expressed as pp’-DDE)/L from mothers living in urban and rural areas, respectively. Once adjusted for 4% of lipid, these contamination values yielded HQ values varying from 0.14 in rural and urban areas to 14.168 for urban area mothers according to the FAO/WHO, and from 0.28 to 28.34 according to IRIS EPA calculations. It is important to highlight that the article was published in the 1980s, the same decade that DDT got banned in Brazil. Matuo et al. [51] analyzed colostrum obtained from 32 mothers living in Ribeirão Preto City, Brazil, in 1983 and 1984. The mean and higher total DDT obtained by the authors presented HQ values (3.5 and 9.52 according to FAO/WHO, and 7 and 19.4 according to IRIS EPA estimation, respectively) higher than one, and represented risk through breast milk consumption, also considering 4% of lipid in the whole milk. Other HQ values from studies around the world are shown in Appendix A. In our study, the calculated HQ points out that milk consumption could not be considered safe for just one infant, and suggests that breastfeeding must be encouraged once breast milk is considered a safe and nutritionally appropriate food for children. Based on the contamination levels of ΣDDT in our analyzed milk samples, breastfeeding benefits outweigh the risks.

Our study observed positive and strong correlations between ΣDDT, ΣOCP, and ΣOC in maternal and cord blood samples. These results were expected, as the summation of OC and OCP included the sum of all DDT metabolites. However, there were no correlations between maternal and cord blood for all organochlorine metabolites that could indicate a substantial transfer of these substances by the placenta; although, studies have shown that the OCs transfer from the mother to the fetus [52,53]. Additionally, no correlations were found between maternal blood and breast milk samples, which could indicate the transfer of such compounds to infants by breastfeeding. However, correlation analyses were performed with a small sample size, and the results can be imprecise. In fact, we recognize the small sample size in our study as a limitation, as it could have affected most of bivariate and multivariate analyses, such as the regression ones.

The OCP and PCB concentrations in plasma samples were reported per unit volume of serum (ng/mL), not per unit volume of serum adjusted for the lipid content, and thus, comparisons of OCP and PCB concentrations were made possible, while other studies were limited to those that used the same unit of concentration.

## 5. Conclusions

Our results showed that pregnant women and their newborns are exposed to several persistent organic pollutants, even after decades of their banishment in Brazil. Although detection rates and concentrations of the majority of OCPs and PCBs were low, it is possible to state that pregnant women and their newborns living in a metropolitan area of Rio de Janeiro City are exposed to these substances, and the knowledge of their impact on childhood development is still a question to be solved.

Due to the potentially harmful effects of these substances on human reproduction and development, exposure to OCPs and PCBs, especially in vulnerable groups, such as pregnant women, women intending to conceive and newborns, need to be investigated in a larger population sample. Future results from the Rio Birth Cohort (PIPA study) certainly will allow us to confirm some associations observed in this pilot study, especially those between OCPs and age, BMI and Apgar score.

## Figures and Tables

**Figure 1 ijerph-20-00778-f001:**
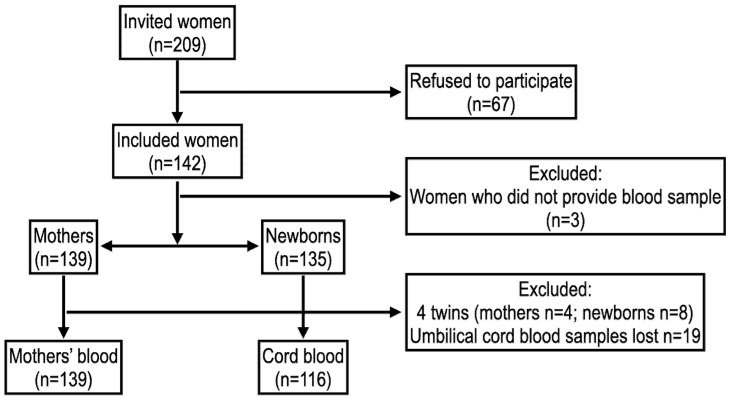
Flow diagram of the study population.

**Table 1 ijerph-20-00778-t001:** Main characteristics of the studied population.

	N	%
**Mothers’**		
**age (years)**		
16–19	12	9.0
20–39	110	82.7
≥40	11	8.3
**Ethnicity**		
White	34	26.4
Non-white	95	73.6
**Income ***		
≤1733.33	36	31.3
1733.34–3000.00	52	45.2
>3000.00	27	23.5
**Schooling**		
High school or less	100	76.3
Higher education	31	23.7
**Body Mass Index**		
<25	55	47.0
25–29.9	39	33.3
≥30	23	19.7
**Alcohol consumption**		
Yes	61	46.6
No	70	53.4
**Smoking**		
Never	103	79.8
Before pregnancy	14	10.9
During pregnancy	12	9.3
**Passive smoking**		
Yes	62	48.4
No	66	51.6
**Newborn sex**		
Male	62	54.9
Female	51	45.1
**Birth weight (g)**		
<2500	8	7.0
≥2500	107	93.0
**Size for gestational age**		
Small for gestational age	8	7.8
Appropriate for gestational age	84	81.5
Large for gestational age	11	10.7
**Gestational age (weeks)**		
<37	10	9.8
≥37	92	90.2
**Apgar 5th min**		
≤8	22	19.5
>8	91	80.5

* Brazilian reais.

**Table 2 ijerph-20-00778-t002:** OCP and PCB concentrations in maternal and umbilical cord blood of the Rio Birth Cohort pilot study. Rio de Janeiro, Brazil, September 2017 to February 2018.

	LOD * ng/mL	Samples > LOD N (%)	GM ** (95%CI)	Min	25%	50%	75%	Max
**Mothers’ blood**								
4,4′-DDE	0.045	30 (22.22)	0.131 (0.089–0.193)	0.050	0.069	0.087	0.198	3.196
∑DDT	-	33 (24.44)	0.136 (0.090–0.204)	0.030	0.067	0.085	0.215	3.249
β-HCH	0.02	12 (8.89)	0.121 (0.105–0.140)	0.090	0.102	0.119	0.144	0.179
γ-HCH	0.07	10 (7.41)	0.175 (0.150–0.205)	0.136	0.145	0.171	0.216	0.251
∑HCH	-	13 (9.63)	0.230 (0.173–0.306)	0.090	0.154	0.248	0.328	0.430
∑PCB ^a^	-	11 (8.15)	0.110 (0.081–0.149)	0.064	0.068	0.097	0.160	0.255
∑OCPs ^a^	-	41 (30.37)	0.187 (0.135–0.259)	0.030	0.080	0.171	0.308	3.249
∑POPs	-	46 (34.07)	0.190 (0.141–0.255)		0.084	0.169	0.305	3.249
**Umbilical cord blood**								
2,4′-DDT	0.02	8 (6.90)	0.133 (0.083–0.215)	0.065	0.082	0.183	0.206	0.254
4,4′-DDD	0.03	9 (7.76)	0.066 (0.037–0.118)	0.030	0.030	0.071	0.122	0.226
4,4′-DDE	0.05	18 (15.52)	0.150 (0.086–0.264)	0.052	0.065	0.110	0.266	5.440
4,4′-DDT	0.05	12 (10.34)	0.099 (0.059–0.168)	0.050	0.053	0.071	0.187	0.477
∑DDT	-	33 (28.45)	0.151 (0.100–0.227)	0.030	0.064	0.100	0.334	6.231
β-HCH	0.02	13 (11.21)	0.071 (0.047–0.105)	0.023	0.038	0.093	0.119	0.169
γ-HCH	0.07	6 (5.17)	0.157 (0.139–0.178)	0.138	0.140	0.157	0.177	0.178
∑HCH	-	15 (12.93)	0.136 (0.082–0.226)	0.023	0.073	0.178	0.289	0.442
PCB 153	0.02	6 (5.17)	0.076 (0.038–0.149)	0.021	0.043	0.080	0.155	0.204
∑PCB ^b^	-	11 (9.48)	0.225 (0.118–0.430)	0.050	0.106	0.193	0.457	1.150
∑OCPs ^b^	-	40 (34.48)	0.177 (0.124–0.252)	0.050	0.074	0.251	0.558	6.220
∑POP	-	41 (35.34)	0.192 (0.130–0.284)	0.050	0.074	0.247	0.734	6.270

* LOD—Detection limit; ** GM—Geometric mean; ^a,b^ ∑OCs and ∑PCBs included some organochlorines and congeners are not shown due to detection rates lower than 5%.

**Table 3 ijerph-20-00778-t003:** Levels of OCPs and PCBs according to maternal sociodemographic characteristics and birth variables. Rio Birth Cohort pilot study. Rio de Janeiro, Brazil, September 2017 to February 2018.

	N; GM * (95%CI)			
	∑DDT	∑HCH	∑OCP	∑POP
**Mothers**				
**Age**				
16–19	2; 0.07 (0.02–0.25)	1; 0.27 (-)	4; 0.18 (0.04–0.70)	4; 0.18 (0.04–0.78)
20–39	22; 0.16 (0.10–0.25)	9; 0.21 (0.14–0.31)	29; 0.18 (0.12–0.26)	34; 0.18 (0.13–0.25)
≥40	9; 0.19 (0.06–0.61)	2; 0.12 (0.13–1.14)	8; 0.23 (0.07–0.69)	8; 0.23 (0.08–0.69)
**Ethnicity**				
White	13; 0.11 (0.07–0.20)	5; 0.29 (0.22–0.39)	15; 0.19 (0.12–0.30)	16; 0.19 (0.13–0.29)
Non-white	19; 0.15 (0.08–0.29)	8; 0.20 (0.13–0.31)	25; 0.19 (0.12–0.31)	29; 0.19 (0.12–0.29)
**Income**				
≤1733.33	4; 0.21 (0.02–2.32)	3; 0.24 (0.18–0.40)	6; 0.27 (0.08–0.89)	7; 0.25 (0.09–0.70)
1733.34–3000.00	10; 0.11 (0.07–0.17)	5; 0.24 (0.18–0.31)	14; 0.17 (0.12–0.24)	18; 0.17 (0.13–0.23)
>3000.00	13; 0.13 (0.06–0.30)	4; 0.19 (0.07–0.52)	14; 0.18 (0.09–0.38)	14; 0.19 (0.09–0.39)
**Body Mass Index**				
<25	11; 0.13 (0.07–0.26)	7; 0.28 (0.19–0.42)	16; 0.18 (0.11–0.30)	18; 0.19 (0.12–0.30)
25–29.9	12; 0.10 (0.05–0.23)	1; 0.28 (-) †	12; 0.14 (0.07–0.31)	13; 0.15 (0.07–0.29)
≥30	8; 0.25 (0.09–0.72)	3; 0.18 (0.02–0.68)	10; 0.30 (0.15–0.60)	10; 0.32 (0.16–0.64)
**Newborns**				
**Sex**				
Male	16; 0.18 (0.11–0.30)	9; 0.11 (0.05–0.23)	21; 0.19 (0.13–0.29)	22; 0.21 (0.13–0.33)
Female	16; 0.13 (0.06–0.27)	5; 0.17 (0.05–0.54)	17; 0.16 (0.08–0.34)	17; 0.18 (0.08–0.40)
**Birth weight (g)**				
<2500	3; 0.08 (0.05–0.13)	1; 0.25 (0.01–0.86)	4; 0.11 (0.04–0.27)	4; 0.11 (0.04–0.27)
≥2500	29; 0.17 (0.10–0.26)	14; 0.13 (0.08–0.22)	35; 0.19 (0.13–0.29)	36; 0.21 (0.14–0.33)
**Gestational age (weeks)**				
<37	4; 0.11 (0.04–0.34)	3; 0.17 (0.02–1.33)	5; 0.16 (0.05–0.55)	5; 0.17 (0.04–0.77)
≥37	23; 0.16 (0.09–0.28)	9; 0.08 (0.01–0.62)	28; 0.19 (0.12–0.30)	29; 0.20 (0.12–0.34)
**Apgar 5th min**				
≤8	6; 0.53 (0.11–2.51)	5; 0.08 (0.02–0.45)	7; 0.56 (0.16–1.98)	8; 0.54 (0.15–1.96)
>8	26; 0.12 (0.08–0.17)	10; 0.17 (0.12–0.25)	32; 0.14 (0.10–0.20)	32; 0.15 (0.11–0.22)

* Geometric Mean; † not a mean (n = 1).

**Table 4 ijerph-20-00778-t004:** Spearman correlation table for detected POPs concentrations in maternal and cord blood and breast milk. Rio Birth Cohort pilot study, Rio de Janeiro, Brazil, September 2017 to February 2018.

		ΣDDT _Mother_	ΣDDT _Cord_	ΣOCP _Mother_	ΣOCP _Cord_	ΣOC _Mother_	ΣOC _Cord_	ΣHCH _Mother_	ΣHCH _Cord_	DDE _Mother_	DDE _Cord_	DDE _BM 1°m_ *	DDE _BM 3°m_ **
ΣDDT _Mother_	ρ	1.000	0.060	0.709	0.021	0.684	0.007	0.500	0.800	1.000	−0.500	0.286	0.257
	p		0.845	<0.0001	0.940	<0.0001	0.980	0.667	0.200		0.667	0.535	0.623
	n	33	13	32	15	32	15	3	4	2	3	7	6
ΣDDT _Cord_	ρ		1.000	−0.006	0.913	−0.029	0.911		0.500	−0.543	−0.700	0.500	−0.400
	p			0.983	<0.0001	0.911	<0.0001		0.667	0.266	0.188	0.667	0.600
	n		33	16	32	16	32	1	3	6	5	3	4
ΣOCP _Mother_	ρ			1.000	−0.051	0.964	−0.061	−0.400	−0.200	−0.543	0.029	0.429	0.517
	p				0.836	<0.0001	0.803	0.600	0.704	0.266	0.957	0.337	0.154
	n			41	19	43	19	4	6	6	6	7	9
ΣOCP _Cord_	ρ				1.000	−0.047	0.996		−0.800	−0.371	−0.500	0.400	−0.300
	p					0.847	<0.0001		0.200	0.468	0.391	0.600	0.624
	n				40	19	38	1	4	6	5	4	5
ΣOC _Mother_	ρ					1.000	−0.056	−0.200	0.071	−0.371	0.257	0.439	0.417
	p						0.819	0.747	0.879	0.468	0.623	0.337	0.265
	n					46	19	5	7	6	6	7	9
ΣOC _Cord_	ρ						1.00		−0.800	−0.393	−0.543	0.400	−0.300
	p								0.200	0.383	0.266	0.600	0.624
	n						40	1	4	7	6	4	5
ΣHCH _Mother_	ρ							1.000	0.500	0.600	1.000	−0.500	0.500
	p								0.667	0.285		0.667	0.667
	n							13	3	5	3	3	3
ΣHCH _Cord_	ρ								1.000	0.551	−0.086		
	p									0.257	0.872		
	n								15	6	6	1	1
DDE _Mother_	ρ									1.000	0.484	0.548	−0.042
	p										0.094	0.160	0.907
	n									30	13	8	10
DDE _Cord_	ρ										1.000	0.500	−0.300
	p											0.667	0.624
	n										18	3	5
DDE _BM 1°m_ *	ρ											1.000	0.429
	p												0.337
	n											19	7
DDE _BM3°m_ **	ρ												1.000
	p												
	n												22

* BM 1°m: breast milk 1st month; ** BM 3°m: breast milk 3rd month.

**Table 5 ijerph-20-00778-t005:** Regression models for maternal blood levels of ΣDDT predictors.

Independent Variables	Coefficient (95%CI)
** *Univariate models* **	
Age	0.037 (−0.012; 0.086)
BMI	0.024 (−0.044; 0.093)
Education (high school or more)	−0.082 (−0.989; 0.823)
Ethnicity (non-white)	0.131 (−0.748; 1.010)
** *Multivariate model 1* **	
Age	0.027 (−0.034; 0.089)
BMI	0.010 (−0.067; 0.086)
** *Multivariate model 2* **	
Age	0.027 (−0.037; 0.092)
BMI	0.010 (−0.070; 0.090)
Ethnicity (non-white)	0.183 (−0.748; 1.113)

Model 1: Dependent variable: ΣDDT levels in maternal blood; Independent variables: age and BMI. Model 2: Dependent variable: ΣDDT levels in maternal blood; Independent variables: age, BMI, and ethnicity.

**Table 6 ijerph-20-00778-t006:** Regression analyses of the association between umbilical blood levels of ΣDDT and birth outcomes.

Dependent Variables	Coefficient (95%CI)
** *Univariate models* **	
Birth weight (grams)	−11.777 (−206.718; 183.164)
Gestational length (weeks)	−0.943 (−3.504; 1.618)
Apgar score 5th min	−0.441 (−0.645; −0.237)
** *Multivariate model 1* **	
Apgar score 5th min	−0.440 (−0.648; −0.233)
** *Multivariate model 2* **	
Apgar score 5th min	−0.405 (−0.651; −0.160)

Model 1: Dependent variable: Apgar score 5th min; Independent variables: ΣDDT levels in umbilical blood and birth weight. Model 2: Dependent variable: Apgar score 5th min; Independent variables: ΣDDT levels in umbilical blood, birth weight and gestational length.

## Data Availability

Data supporting the reported results are stored with the corresponding author and is available upon request.

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
