# Peer review of "Persistent Organic Pollutant Levels in Maternal and Cord Blood Plasma and Breast Milk: Results from the Rio Birth Cohort Pilot Study of Environmental Exposure and Childhood Development (PIPA Study)"

_ijerph, 2022, doi:10.3390/ijerph20010778_

Round 1
Reviewer 1 Report
Manuscript Number: ijerph-2050766
Title: Persistent organic pollutant levels in maternal, cord blood plasma and breast milk: Results from the Rio Birth Cohort Pilot Study of Environmental Exposure and Childhood Development (PIPA Study)
The paper measured PCB and OCP concentrations in breast milk, maternal and umbilical cord blood of pregnant women and their newborns in Brazil. However, I don’t think the paper is sufficient to warrant publication for the following reasons:
1. I have one major concern about the data of OCP and PCB, since only 6.9-35% of the samples had OCP and PCB >LOD. I’d like to see more QA/QC in this paper.
2. No related data about daily intake and HQ values are shown in the Results.
3. Those figures and tables are showing the same content.
4. Table 3: Some 95%CI values are missing.
5. Figure 2: Weird to have legend at the bottom of the Figure, please revise.
Author Response
Response to Reviewer 1:
We thank reviewer 1 for their important considerations in improving the manuscript. The responses were highlighted in red in this document and the manuscript.
The paper measured PCB and OCP concentrations in breast milk, maternal and umbilical cord blood of pregnant women and their newborns in Brazil. However, I don’t think the paper is sufficient to warrant publication for the following reasons:
- I have one major concern about the data of OCP and PCB, since only 6.9-35% of the samples had OCP and PCB >LOD. I’d like to see more QA/QC in this paper.
Response: We agree with the reviewer and added more information on QA/QC in the method section as below (page 4, lines136-169):
Spiked plasma were used to prepare calibration curves. Validation parameters (Table S1) included linearity, sensitivity, recovery tests, repeatability, the limit of detection (LOD), and limit of quantification (LOQ), based on the EPA 8081B method [19] and the Brazilian INMETRO Guide [20]. Eight-point calibration curves from 0.2 to 15 ng mL-1 were used to prepare the calibration curves by linear regression. Correlation coefficients ranged from 0.936 to 0.994, showing good linearity for both OCPs pesticides and PCBs. Sensitivity was calculated using the curve slope and varied from 111.655 to 212.082 for DDT and its isomers. The mean percentage recoveries ranged from 93% to 105% for OCP and from 87% to 107% for PCB, evaluated at 0.5, 5, and 15 ng mL-1. LOD and LOQ were calculated from standard deviation multiplied by factor t (Student t) or per 10, respectively, for 7 replicates at 0.2 ng mL-1 lowest level. The factor t for 6 degrees of freedom at a 95% confidence interval was 1.943. The LOD was 0.015 ng mL-1 to 0.468 ng mL-1 for OCP and 0.2 ng mL-1 to 0.36 ng mL-1 for PCB. The LOQ was 0.045 ng mL-1 to 1.419 ng mL-1 for OCP pesticides and 0.05 ng mL-1 to 1.08 ng mL-1 for PCB. Repeatability at 2 ng mL-1 level ranged from 0.5 to 14.9% for within-day variability and from 0.8 to 8.6% for day-to-day variability.
To perform only one analysis per sample and obtain one chromatogram and SRM spectrum, analytical quality controls were performed for each batch using previously spiked plasma at 3 levels, 0.2, 0.5, and 1 ng mL-1, and blank. The control of sample results was made by monitoring the daily abundance values in spiked plasma at these 3 concentration levels and the slope for each batch control. If the variation between the last batches was greater than 20%, the batch was redone. Also, variations were <20% for two randomized samples, with extra available volume, done in duplicate in each batch.
The internal standard was added in all samples and controls to evaluate the injection and chromatographic batch conditions, being monitored over time by its area and all analytes in a standard mixture with greater purity than 99%, at 1 µg mL-1 level. Solvent injections were also made every 10 samples to evaluate the carry-over effect and cleanness of the chromatographic system.
Three Selective Reaction Monitoring transitions were exhaustively checked for each positive sample to avoid false positive results. The relative abundances of the three selected precursor ion-product ion transitions were obtained from high purity standards and monitored for each pesticide. Also, a very low variation in retention times was observed.
Briefly, each batch had its blank, quality control samples, and a short calibration curve in spiked plasma, and sample levels were calculated by external standardization using linear regression.
- No related data about daily intake and HQ values are shown in the Results.
Response: We agree with the reviewer and added more information on QA/QC in the result section as below (page 9, lines 287-290):
The calculated EDI varied from 5.98x10-6 to 1.17x10-3 μg of ΣDDT/kg body wt./day. Mean EDI 0.21μg of ΣDDT/kg body wt./day. The HQ values ranged from 5.98x10-3 and 1.20x10-2 to 1,17 and 2,34 according to EPA and FAO/WHO respectively, and just one milk sample yielded HQ value higher than 1.
We also added in the discussion section (page 12, lines 414-428):
DDT residues and HQ have been calculated in breast milk samples in Brazilian studies. For example, Sant'ana et al. [51] analyzed human milk samples collected from rural and urban areas in Botucatu City, Brazil. The values expressed in whole milk varied from 1.0 to 101.2 µg of total DDT (expressed as op’-DDT+pp’-DDT+pp’-DDD)/L and from 1.0 to 62.1µg of total DDT (expressed as pp’-DDE)/L from mothers living in urban and rural areas respectively. Once adjusted for 4% of lipid, these contamination values yielded HQ values varying from 0.14 in rural and urban areas to 14.168 for urban area mothers according to the FAO/WHO and from 0.28 to 28.34 according to IRIS EPA calculations. It is important to highlight that the article was published in the 1980s, the same decade that DDT got banned in Brazil. Matuo et al. [52] analyzed colostrum obtained from 32 mothers living in Ribeirão Preto City, Brazil, in 1983 and 1984. The mean and higher total DDT obtained by the authors presented HQ values (3.5 and 9.52 according to FAO/WHO, and 7 and 19.4 according to IRIS EPA estimation, respectively) higher than one and represented risk through breast milk consumption, also considering 4% of lipid in the whole milk. Other HQ values from studies around the world are shown in Table S2.
- Those figures and tables are showing the same content.
Response: We excluded the figures according to the reviewer’s suggestion.
- Table 3: Some 95%CI values are missing.
Response: We included the missing 95%IC in Table 3.
- Figure 2: Weird to have a legend at the bottom of the Figure, please revise.
Response: We excluded the figures and the legend.

Reviewer 2 Report
Thank you for allowing me to review this article for the IJERPH magazine. The study is very interesting trying to stablish the level of persistent organic pollutants in cord samples and blood. However, I am concern about stadistical analysis is a bit poor for a Q1 journal (they only use descriptive analysis). I suggest the authors resubmit the article after reviewing this aspect, including for example a correlation study between the different kind of samples and a multivariate analysis that adds additional value to a simple descriptive analysis. I am convinced that the study has greater potential than what the authors have extracted.
Author Response
RESPONSE TO REVIEWER 2
We are grateful for your consideration and suggestions, which have been very helpful in improving the manuscript. The responses were highlighted in red in this document and the manuscript.
Thank you for allowing me to review this article for the IJERPH magazine. The study is very interesting trying to establish the level of persistent organic pollutants in cord samples and blood. However, I am concern about statistical analysis is a bit poor for a Q1 journal (they only use descriptive analysis). I suggest the authors resubmit the article after reviewing this aspect, including for example a correlation study between the different kind of samples and a multivariate analysis that adds additional value to a simple descriptive analysis. I am convinced that the study has greater potential than what the authors have extracted.
Response: We agree with the reviewer’s suggestion and ran the Spearman correlation between DDE, ΣHCH, ΣDDT, ΣOCP, and ΣOC in the maternal and cord blood and breast milk during the 1° and 3° months of follow-up. The results were included on pages 9-10, and a discussion was added on pages 12-13.

Round 2
Reviewer 1 Report
Even though the authors have provided QA/QCs and revised the paper as suggested, I still have some concerns about the results presented. In the revised manuscript, the authors added a new Table about correlations between POPs concentrations. For some of the correlation analysis, there are only 2 or 3 samples, which is meaningless in terms of statistics. I think the topic in this paper is interesting, but a lot of work needs to be done to improve the overall quality, especially the data analysis and discussion.
Author Response
Response to Reviewer 1: We agree with the reviewers that the correlations between POPs concentrations are unsuitable due to the small n. This limitation does not allow us to carry out more sophisticated statistical analyses. The article describes the exposure to organochlorines experienced by pregnant women and their newborns and measured in maternal blood, cord blood, and breast milk. To reviewers, the manuscript in its current format may not indicate a substantial contribution to the journal field. We respectfully disagree, as our study is the first to show that pregnant women and their newborns in Brazil are contaminated with several POPs, even after almost 3 decades of no legal use of these substances in Brazil. These results certainly raise questions about the possible impacts of this exposure on the mothers' and newborns' health. In that sense, IJERPH could benefit from the upcoming results of POPs contamination in the PIPA Study main cohort and its impact on the population's health.
Reviewer 2 Report
Thank you for your effort in revising the original article. I could see how the authors have included a table with the correlation between POPs. However, I was surprised to see the n so small in some metabolites (maybe that should be indicated as a limitation of the article?). Also, no multivariate analysis was included. I still think the article has more potential than the authors have extracted and that the paper's quality must improve. I encourage the authors to make an effort in this regard so that the result is of high quality.
Author Response
Response to Reviewer 2: We agree with the reviewers that the correlations between POPs concentrations are unsuitable due to the small n. This limitation does not allow us to carry out more sophisticated statistical analyses. The article describes the exposure to organochlorines experienced by pregnant women and their newborns and measured in maternal blood, cord blood, and breast milk. To reviewers, the manuscript in its current format may not indicate a substantial contribution to the journal field. We respectfully disagree, as our study is the first to show that pregnant women and their newborns in Brazil are contaminated with several POPs, even after almost 3 decades of no legal use of these substances in Brazil. These results certainly raise questions about the possible impacts of this exposure on the mothers' and newborns' health. In that sense, IJERPH could benefit from the upcoming results of POPs contamination in the PIPA Study main cohort and its impact on the population's health.